

# Diversity and abundance of conspicuous macrocrustaceans on coral reefs differing in level of degradation

Roberto González-Gómez[1,2], Patricia Briones-Fourzán[1], Lorenzo Álvarez-Filip[1] and Enrique Lozano-Álvarez[1]

[1] Instituto de Ciencias del Mar y Limnología, Unidad Académica de Sistemas Arrecifales, Universidad Nacional Autónoma de México, Puerto Morelos, Quintana Roo, Mexico
[2] Posgrado en Ciencias del Mar y Limnología, Universidad Nacional Autónoma de México, Mexico, Ciudad de México, Mexico

## ABSTRACT

Coral reefs sustain abundant and diverse macrocrustaceans that perform multiple ecological roles, but coral reefs are undergoing massive degradation that may be driving changes in the species composition and abundance of reef-associated macrocrustaceans. To provide insight into this issue, we used non-destructive visual census techniques to compare the diversity and abundance of conspicuous macrocrustaceans (i.e., those >1 cm and visible without disturbance) between two shallow Caribbean coral reefs similar in size (∼1.5 km in length) and close to each other, but one ("Limones") characterized by extensive stands of the branching coral *Acropora palmata*, and the other ("Bonanza") dominated by macroalgae and relic coral skeletons and rubble (i.e., degraded). We also assessed the structural complexity of each reef and the percent cover of various benthic community components. Given the type of growth of *A. palmata*, we expected to find a greater structural complexity, a higher cover of live coral, and a lower cover of macroalgae on Limones, and hence a more diverse and abundant macrocrustacean community on this reef compared with Bonanza. Overall, we identified 63 macrocrustacean species (61 Decapoda and two Stomatopoda). Contrary to our expectations, structural complexity did not differ significantly between the back-reef zones of these reefs but varied more broadly on Limones, and the diversity and abundance of macrocrustaceans were higher on Bonanza than on Limones despite live coral cover being higher on Limones and macroalgal cover higher on Bonanza. However, the use of various types of microhabitats by macrocrustaceans differed substantially between reefs. On both reefs, the dominant species were the clinging crab *Mithraculus coryphe* and the hermit crab *Calcinus tibicen*, but the former was more abundant on Bonanza and the latter on Limones. *M. coryphe* occupied a diverse array of microhabitats but mostly coral rubble and relic skeletons, whereas *C. tibicen* was often, but not always, found associated with colonies of *Millepora* spp. A small commensal crab of *A. palmata*, *Domecia acanthophora*, was far more abundant on Limones, emerging as the main discriminant species between reefs. Our results suggest that local diversity and abundance of reef-associated macrocrustaceans are partially modulated by habitat degradation, the diversity of microhabitat types, and the establishment of different commensal associations rather than by structural complexity alone.

Corresponding author
Patricia Briones-Fourzán,
briones@cmarl.unam.mx

## INTRODUCTION

Habitat complexity is an important factor driving the abundance and diversity of associated species by facilitating niche separation and resource partitioning (*Vytopil & Willis, 2001*; *Idjadi & Edmunds, 2006*). Keystone structures (sensu *Tews et al., 2004*) are distinct spatial structures that create complex habitats that facilitate species' coexistence by offering food resources and shelter against predators and various environmental stressors (*Bruno & Bertness, 2001*; *Kerry & Bellwood, 2015*). In coral reefs, keystone structures are created by scleractinian corals, which provide great spatial complexity to the system and multiple shelters for other organisms in the form of crevices, holes, and branches.

The role of corals in maintaining abundant and diverse communities of reef invertebrates is well recognized. For example, *Stella et al. (2011)* identified 869 coral-associated invertebrate species, with arthropods (mostly crustaceans) as the major contributors to the overall diversity. Not only are reef macrocrustaceans (in particular Decapoda and Stomatopoda) highly diverse; they are also abundant and perform multiple ecological roles. They are part of numerous feeding guilds, acting as predators, parasites, herbivores, scavengers, and detritivores, as well as suspension and deposit feeders (*Abele, 1976*; *Glynn & Enochs, 2011*; *Stella et al., 2011*), and constitute a critical link between primary production and a wide array of higher order consumers, including reef fishes (*Randall, 1967*). Some macrocrustaceans defend live coral from potential predators (*McKeon & Moore, 2014*), maintain coral health by clearing sediments (*Stewart et al., 2006*), or eliminate parasites from reef fishes, many of which are of economic value (*Becker & Grutter, 2004*). Therefore, macrocrustaceans are a key component of coral reef ecosystems, making it necessary to understand the potential effects that coral reef degradation may have on their communities.

Coral reefs are undergoing massive degradation due to the effects of multiple stressors, including climate change induced-bleaching, increases in disease outbreaks and prevalence, eutrophication, and invasive or destructive fishing practices (*Hughes et al., 2017*). This is particularly true for Caribbean coral reefs, where declines in reef architectural complexity and phase shifts from coral to macroalgal dominance have been extensively documented (*Gardner et al., 2003*; *Álvarez Filip et al., 2009*; *Bruno et al., 2014*; *Jackson et al., 2014*; *Suchley, McField & Álvarez Filip, 2016*). Coral reef degradation is likely to have serious consequences for ecosystem functioning and services, as well as for reef biodiversity (*Álvarez Filip et al., 2009*), and there is support for this latter assumption in the case of reef fishes (e.g., *Graham et al., 2011*; *Coker, Wilson & Pratchett, 2014*; *Álvarez Filip et al., 2015*; *Newman et al., 2015*). However, predictions for invertebrate taxa are less clear because different studies have reported contrasting results (see *Graham & Nash, 2013*). For example, in Papua New Guinea, the density and abundance of several macroinvertebrate groups, including motile crustaceans, decreased in reefs with lower architectural complexity due to acidification compared with more complex reefs (*Fabricius et al., 2014*). In the US

Virgin Islands, diversity, but not abundance, of invertebrates was positively related with topographic complexity, but not with coral diversity or live coral cover (*Idjadi & Edmunds, 2006*). In contrast, invertebrate assemblages were more diverse and abundant on dead than on live coral habitats in Panama (*Nelson, Kuempel & Altieri, 2016*), whereas habitat complexity accounted for very little of the variability in invertebrates (including arthropods) on Caribbean *Orbicella* reefs (*Newman et al., 2015*). These contrasting results suggest that many reef-associated invertebrates do not necessarily benefit from the presence of live corals per se, but from the complex 3-D framework of coral reefs, which can persist for years after the death of corals.

One of the most structurally complex reef-building corals in the Caribbean region is the branching coral *A. palmata.* This species, once dominant in the region, forms thick stands that provide an intricate network of crevices on shallow-water reefs (<5 m). The populations of this species have sustained extensive mortality since the early 1980s, substantially reducing coral cover, increasing substratum for algal growth, and drastically reducing reef complexity (*Aronson & Precht, 2001*; *Álvarez Filip et al., 2009*; *Jackson et al., 2014*). Currently, *Acropora*-dominated reefs are rare. For example, a recent assessment in 107 sites along the Mesoamerican Reef (MAR) revealed that *A. palmata* was present (mostly at low cover values) in only 20% of the sites, and that only one site ("Limones" reef), located in the northernmost Mexican portion of the MAR, exhibited extensive stands of *A. palmata* resulting in a high (>35%) cover of this branching coral (*Rodríguez-Martínez et al., 2014*). Therefore, assessing the composition and structure of the ecological condition on these *Acropora*-dominated reefs is crucial to elucidate how reef degradation is modifying the ecological relationships on coral reefs.

The aims of the present study were twofold: to obtain a list of species of conspicuous macrocrustaceans associated to shallow Caribbean reef habitats, and to compare the diversity and abundance of these macrocrustaceans between Limones reef and another reef ("Bonanza"), similar in size to Limones but highly degraded, to elucidate the potential effects of coral reef degradation on this type of invertebrates. We predicted that the diversity and abundance of macrocrustaceans would be higher in Limones because the presence of extensive stands of live *A. palmata* would presumably confer a greater structural complexity and provide a broad diversity of microhabitats potentially used by reef-associated species (*Roberts & Ormond, 1987*; *Garpe et al., 2006*).

## MATERIALS AND METHODS

### Study site

The study was conducted at the Puerto Morelos Reef National Park (PMRNP; Fig. 1A), a marine protected area located on the NE coast of the Yucatan Peninsula, Mexico. The PMRNP is an extended fringing reef system composed of a series of reef units that differ in size and structural complexity (*Lozano-Álvarez et al., 2017*; *Morillo-Velarde et al., 2018*), separated from the coast by a shallow (<5 m) reef lagoon. Along the reef tract, coral cover tends to be greater on the back-reef and crest zones than on the low-relief fore-reef, which descends gradually into an extensive sand platform at 20–25 m (*Jordán-Dahlgren, 1993*).

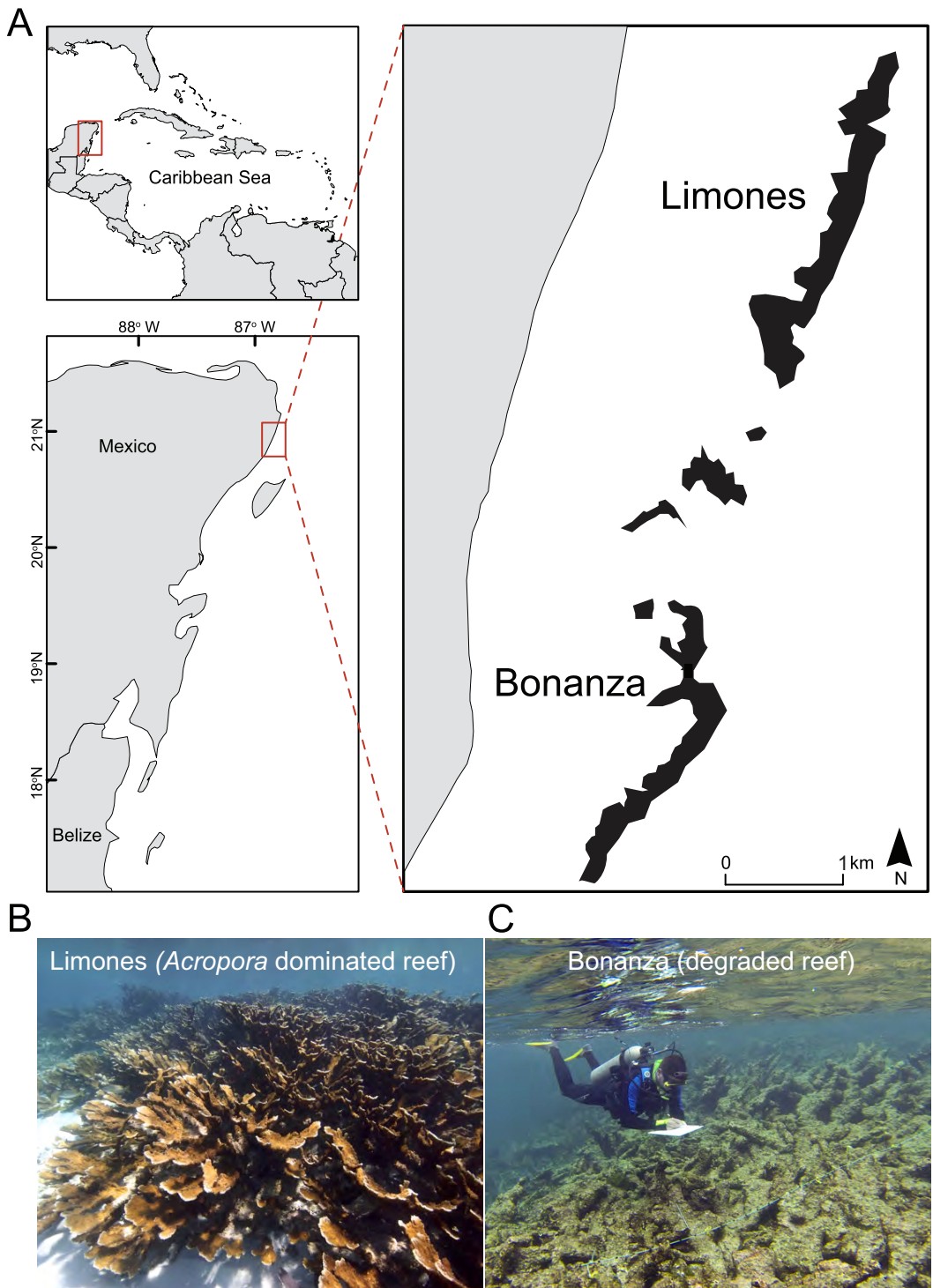

**Figure 1  Study area.** (A) Location of the studied reef units, Limones (well-preserved) and Bonanza (degraded), at Puerto Morelos, México, and photographs showing the current state of (B) Limones and (C) Bonanza (Photo credits B: Lorenzo Álvarez-Filip; C: Fernando Negrete-Soto).

Two of these reef units are Limones (centered at 20°59.1′N, 86°47.9′W) and Bonanza (centered at 20°57.6′N, 86°48.9′W) (Fig. 1A). Both reefs are similar in length (~1.5 km), depth and distance from the coast, but differ in their level of degradation, as indicated by several studies. Healthy and resilient populations of *A. palmata* have been reported since 1985 on Limones (*Rodríguez-Martínez et al., 2014*) (Fig. 1B), and recently *Morillo-Velarde et al. (2018)* found 50% live coral cover, mostly *A. palmata*, along the central part of Limones. In contrast, live coral cover on Bonanza has gradually declined from 33% in 1985 (*Jordán-Dahlgren, 1993*) to 12% in 2006–2007 (*Carriquiry et al., 2013*) and 7% by 2015, when it exhibited extensive areas of relic *Acropora* skeletons (Fig. 1C) and a predominance of erect macroalgae (>60% cover) (*Morillo-Velarde et al., 2018*). Based on a number of broad- and local-scale resilience indicators, including coral cover, *Ladd & Collado-Vides (2013)* categorized Limones as a high-resilience site and Bonanza as a low-resilience site, whereas based on two different reef health indices, *Díaz-Pérez et al. (2016)* categorized the health of Bonanza as "poor". Fishing activities have been banned on both Limones and Bonanza reefs since 1996. However, Bonanza is open to visitation, whereas tourist activities are not allowed on Limones since 2014 given the high ecological value of this reef (*Rodríguez-Martínez et al., 2014*).

## Macrocrustacean surveys

Sampling by divers remains the most efficient way to find reef-associated species when they are large enough to be seen (*Knowlton et al., 2010*; *Giraldes, Coelho Filho & Coelho, 2012*). Therefore, we used SCUBA diving to conduct quantitative surveys of conspicuous macrocrustaceans (herein defined as motile crustaceans larger than ~1 cm) via belt transects, with a permit issued by Comisión Nacional de Acuacultura y Pesca (PPF/DGOPA-259/14). All underwater samplings were conducted by two scientific observers, who were thoroughly trained in macrocrustacean identification over several months prior to the samplings. Training was achieved by repeatedly studying an extensive guide of local crustacean species created in our lab with photos from many different sources, followed by direct identification in the field during several preliminary dives. In all cases, the results were cross-checked between both divers (*Lessios, 1996*; *Backus, 2007*). On each reef, we haphazardly laid 30 25-m transects on the back reef zone along the length of the reef. The two divers recorded all macrocrustaceans observed within 1 m to the right and 1 m to the left of the transect line (i.e., an area of 50 m$^2$ per transect), both over the substrata and under coral rubble. Individuals were identified in situ to the lowest possible taxonomic level and many were extensively photographed underwater to further help in their identification. Only a few individuals were collected in zip-lock bags and taken to the laboratory for their identification. Also recorded was the type of microhabitat in which each specimen was observed. These microhabitats included *A. palmata*, *Agaricia agaricites*, other live corals, *Millepora* spp., dead coral skeletons, coral rubble, gorgonians, algae, anemones, and sand. Despite their relatively large size, many macrocrustaceans hide deeply in holes and crevices in coral reefs during the day but forage over the reef substrata at night; therefore, to obtain a species list as complete as possible, we further conducted qualitative surveys on each reef by recording all species observed during three separate nocturnal 1-h dives.

## Structural complexity and benthic community

We assessed the current ecological condition of the back-reef zones of Limones and Bonanza by using two metrics of structural complexity and estimating the percent cover of different components of the benthic community. Structural complexity was assessed with the Habitat Assessment Score (HAS; *Gratwicke & Speight, 2005*), which is a qualitative metric, and the rugosity index, which is a quantitative metric (*Risk, 1972*; *Álvarez Filip et al., 2009*). HAS provides an overall structural complexity value by visually evaluating six variables of the local topography (rugosity, variety of growth forms, height, refuge size categories, percentage of live cover, and percentage of hard substratum). Each variable is assigned a score between 1 and 5 (from smallest or lowest to largest or highest), and the sum of the individual scores is the HAS. Therefore, a score of 6 would represent the least complex habitats and a score of 30 would represent the most complex habitats. HAS values were obtained in three 4 m$^2$ quadrats positioned at the beginning, middle, and end of nine of 25-m transects per reef, and the three values were averaged to obtain the transect-level HAS.

Rugosity is the ratio of the length of a chain molded to the reef surface to the linear distance between its start and end points. A perfectly flat surface would have a rugosity index of 1, with larger numbers indicating more complex surfaces (*Risk, 1972*). To measure rugosity, a chain (0.5 cm link-length) was molded to the reef surface along 24 10-m long transects on Limones and 21 on Bonanza. These 10-m transects were also used to estimate percent cover of components of the benthic community via the point intercept method (*Hill & Wilkinson, 2004*). The transects were marked every 10 cm, thus yielding 100 points per transect (*Lang et al., 2010*), A diver recorded which of the following benthic components was found under each mark: live hard corals, calcareous macroalgae, fleshy macroalgae, coralline algae, algal turf, cyanobacterial mat, other invertebrates (e.g., zoanthids, *Millepora*, *Cliona*), and other components (e.g., sand, seagrass).

## Data analysis

### Structural complexity and benthic community

HAS values and rugosity indices were compared between reefs with Mann–Whitney *U* tests. A significance level of 95% was used in all cases. The percent data on the benthic community structure were logit-transformed (*Warton & Hui, 2011*) and subjected to a principal component analysis (PCA). Then, the transformed data for each benthic component was compared between reefs with a Student's *t*-test.

### Characterization of the macrocrustacean community

Quantifying biodiversity is problematic because there is no single "best" index. However, simple indices (i.e., those that measure species richness) can be slightly preferable when the primary goal is to detect effects of external factors on diversity, whereas compound indices (i.e., those that combine measures of richness and abundance) can be preferable when the primary goal is to differentiate sites by their level of diversity (*Magurran & Dornelas, 2010*; *Morris et al., 2014*). Given the aims of the present study, we estimated both types of indices for the macrocrustaceans from each reef. These indices included species richness (*S*, number of species), Simpson's dominance ($D = \sum (n_i/N)^2$, where $n_i$ is

number of individuals of the $i$th species and $N$ is total number of individuals), Shannon–Wiener's diversity ($H' = -\sum_{i=1}^{s} p_i \log 2 p_i$, where $H'$ is the information contained in the sample (bits/individual) and $p_i = n_i/N$), and Pielou's evenness ($J' = H'/\log S$). Each index was compared between reefs with a Mann–Whitney $U$ test. Species accumulation and rarefaction curves were computed using EstimateS v9.1.0 (*Gotelli & Colwell, 2001*).

The community composition of macrocrustaceans was analyzed using multivariate techniques with PRIMER 6 v6.1.9 (PRIMER-E Ltd). Differences in the taxonomic composition between Limones and Bonanza were analyzed by non-metric multidimensional scaling (MDS) on fourth-root transformed data, using the Bray-Curtis similarity measure (*Clarke, 1993*). The statistical significance of the observed differences in the macrocrustacean assemblages between reefs was further tested with a one-way analysis of similarity (ANOSIM). This test provides an $R$-value indicative of the degree of difference between samples as well as a $p$-value for the significance of that difference. $R$ values close to 0 are indicative of little difference while values close to 1 are indicative of a large difference in sample composition (*Clarke & Warwick, 2001*). Finally, we did a similarity percentage analysis (SIMPER, *Clarke, 1993*) to identify those species responsible for the observed differences in community composition between both reefs. For each of the 10 most abundant species, we also compared the density, standardized as individuals (ind.) 50 m$^{-2}$, between reefs with individual Student's $t$ tests.

## RESULTS

### Structural complexity and benthic community components

The median rugosity over the back-reef zones of Limones (1.33 [1.13–2.21], median [interquartile range]) and Bonanza (1.24 [1.07–1.38]) did not differ significantly (Mann–Whitney U test, $U = 182$, $z = 1.764$, $n_1 = 25$, $n_2 = 21$, $p = 0.078$) (Fig. 2A). A similar result was obtained for the median HAS (Fig. 2B) (Limones: 18 [16–20], Bonanza: 19 [17–19]; Mann–Whitney $U$ test, $U = 38.5$, $z = 0.133$, $n_1 = n_2 = 9$, $p = 0.895$). However, the range in values of both metrics, in particular rugosity (Fig. 2A), was substantially broader for Limones (1.05–3.56) than for Bonanza (1.02–2.2), with rugosity values $\geq 2$ obtained on 32% transects on Limones versus 9.5% transects on Bonanza. The percent cover of various components of the benthic community differed between reefs (Fig. 3). In particular, live coral cover was much higher on Limones, whereas the cover of fleshy macroalgae, calcareous macroalgae, and cyanobacterial mats was significantly higher on Bonanza (Fig. 3). In the PCA, the first two components explained 63% of the total variance (Fig. 4). The first component explained 40.2% of the variance and was positively correlated with fleshy macroalgae (loading: 0.640) and negatively correlated with live hard coral ($-0.685$). The second component explained 22.8% of the variance and was strongly positively correlated with turf algae (0.728) and negatively correlated with live hard coral ($-0.449$) (Fig. 4). Most transects on Limones differed from those on Bonanza along the first component.
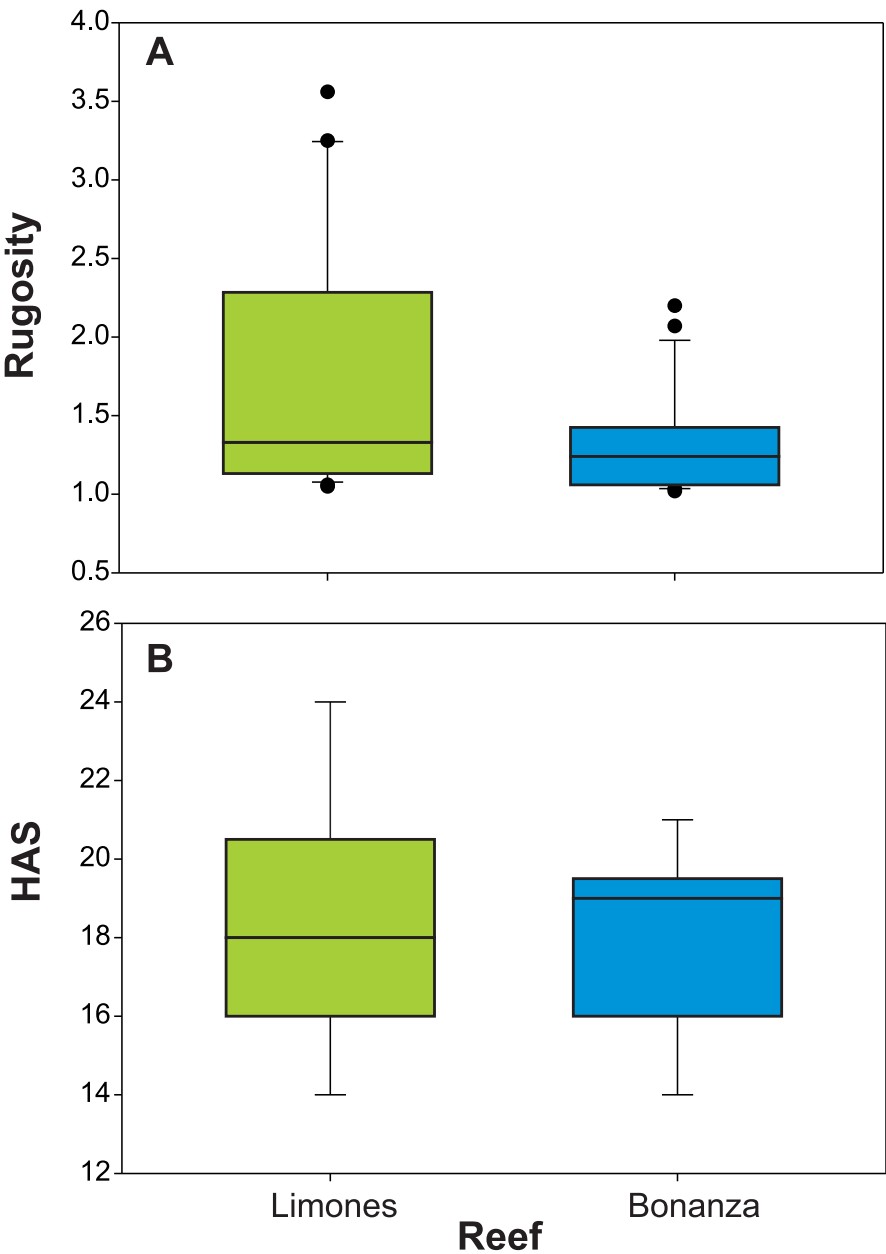

**Figure 2** **Metrics of reef structural complexity.** Box plots of (A) rugosity index and (B) habitat assessment score (HAS) on Limones (green boxes) and Bonanza reefs (blue boxes). The lower and higher boundaries of the box indicate the 25th and 75th percentiles, respectively. The horizontal line within the box marks the median. Whiskers (error bars) above and below the box indicate the 90th and 10th percentiles. Black dots denote outliers.

## The macrocrustacean assemblage

In all, we registered 63 species of macrocrustaceans (Table 1), including six that were only observed during the nocturnal dives (i.e., not quantified). These species were representatives of the Infraorders Brachyura (33 species), Caridea (10), Anomura (10), Achelata (five),

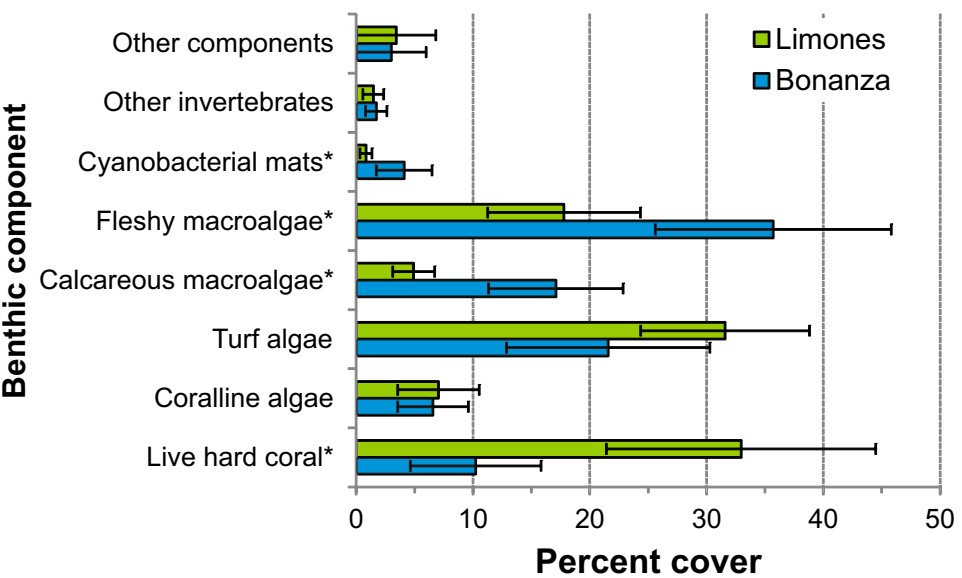

**Figure 3** **Percent cover of benthic community components.** Percent cover of different benthic community components over Limones reef (green columns) and Bonanza reef (blue columns). Error bars are 95% confidence intervals. Asterisks at the end of a component name denote significant differences between reefs ($\alpha = 0.05$).

Axiidea (two), and Stenopodidea (one); the Superfamily Penaeoidea (one), and the stomatopod family Gonodactyloidea (two species). Twelve species, mostly rare (i.e., with <three individuals), could only be identified to the superfamily or family level. The most diverse superfamily was Majoidea, with 22 species. The number of species was higher on Bonanza than on Limones (43 vs 33 species), as was the abundance (2800 vs 2067 individuals) (Table 1).

On both reefs, the number of species increased with the number of transects (accumulation curves, Fig. 5), but more steeply on Bonanza than on Limones. Rarefaction curves did not reach an asymptote for either reef, suggesting that the species richness of conspicuous macrocrustaceans on these reefs is even higher. All ecological indices differed significantly between reefs (Table 2), with Bonanza exhibiting higher levels of species richness ($S$, Mann–Whitney $U$ test, $U = 265.5$, $z = -2.741$, $n_1 = n_2 = 30$, $p = 0.004$), diversity ($H'$, $U = 208$, $z = -3.57$, $n_1 = n_2 = 30$, $p = 0.0002$), and evenness ($J'$, $U = 261$, $z = -2.787$, $n_1 = n_2 = 30$, $p = 0.003$), whereas dominance was higher at Limones ($D$, $U = 176$, $z = -4.044$, $n_1 = n_2 = 30$, $p < 0.0001$).

Macrocrustacean assemblages differed significantly between reefs (ANOSIM, $R = 0.279$, $p < 0.001$) but with some overlap (Fig. 6), suggesting a similar abundance of some species on both reefs. Indeed, SIMPER revealed that the clinging crab *Mithraculus coryphe* and the hermit crab *C. tibicen* were the most abundant species on both reefs, accounting for 71.4% and 62.2% of the similarities observed in Limones and Bonanza, respectively (Table 3). Within Limones, the composition of macrocrustaceans exhibited an average similarity among transects of 48.1%, mainly due to three species: *C. tibicen*, *M. coryphe*

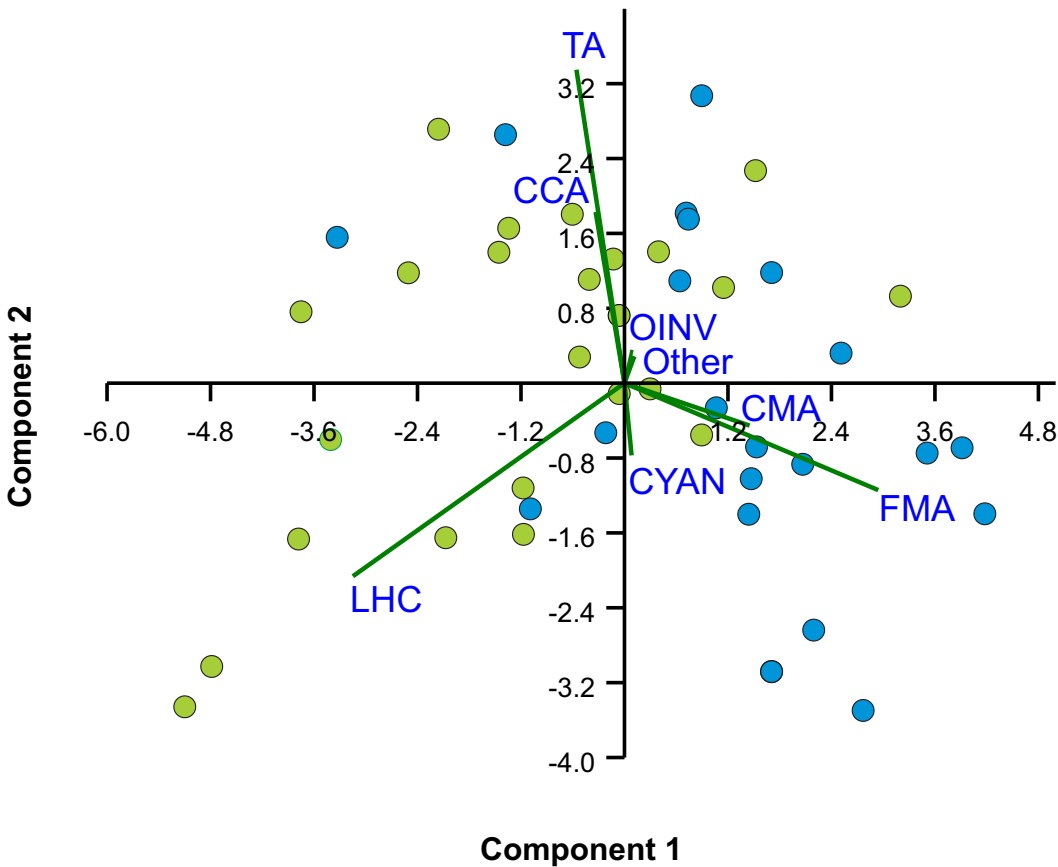

**Figure 4  Principal Components Analysis (PCA) of percent cover of benthic components.** Bi-plot on logit-transformation of percent cover of benthic components over the two studied reefs, Limones (green dots) and Bonanza (blue dots). Each dot represents a transect. LHC, live hard coral; TA, turf algae; FMA, fleshy macroalgae; CMA, calcareous macroalgae; CCA, coralline algae; CYAN, cyanobacterial mats; OINV, other sessile invertebrates; Other, other components (sand, seagrass).

and *D. acanthophora*, with *C. tibicen* as the major contributor (45.2%). Within Bonanza, the average similarity among transects was 46.9%, with *M. coryphe* emerging as the main contributor (33.1%), followed by *C. tibicen* (29.1%) and *Neogonodactylus oerstedii* (8.8%). On Limones, six species accounted for 90% of the observed similarity, whereas on Bonanza, this same percentage was accounted for by eight species (Table 3). The crustacean assemblages of Limones and Bonanza exhibited a mean dissimilarity of 58.5%, with *D. acanthophora* as the main contributor to this dissimilarity (8.1%), followed by *M. coryphe* (6.6%) and *Pagurus brevidactylus* (6.4%).

Despite wide variability among transects, the density of some of the most abundant species differed significantly between reefs (Fig. 7). This was the case for *D. acanthophora*, which had a significantly higher density on Limones (12.6 ± 7.8 ind. 50 m$^{-2}$, mean ± 95% CI, than on Bonanza (1.5 ± 1.9 ind. 50 m$^{-2}$), and for *M. coryphe*, which exhibited a higher density on Bonanza (33.9 ± 14.4 ind. 50 m$^{-2}$) than on Limones (9.7 ± 3.6 ind. 50 m$^{-2}$), as was also the case for *M. sculptus* and *N. oerstedii* (Fig. 7). In contrast, the density

**Table 1  Macrocrustacean species by reef.** Number of macrocrustacean species and individuals registered by visual census ($n = 30$ transects). Bonanza reef: 43 species; 2,800 individuals; Limones reef: 33 species; 2,067 individuals. An X denotes that a species was only qualitatively recorded during nocturnal dives.

|  | Species | Bonanza | Limones |
|---|---|---|---|
| 1 | *Mithraculus coryphe* (Herbst, 1801) | 1,017 | 290 |
| 2 | *Calcinus tibicen* (Herbst, 1791) | 1,002 | 1,143 |
| 3 | *Pagurus brevidactylus* (Stimpson, 1859) | 97 | 48 |
| 4 | *Teleophrys ruber* (Stimpson, 1871) | 95 | 40 |
| 5 | *Paguristes tortugae* Schmitt, 1833 | 84 | 0 |
| 6 | *Mithraculus sculptus* (Lamarck, 1818) | 70 | 17 |
| 7 | *Paguristes anomalus* Bouvier, 1918 | 66 | 15 |
| 8 | *Neogonodactylus oerstedii* (Hansen, 1895) | 57 | 15 |
| 9 | *Domecia acanthophora* (Desbonne in Desbonne & Schramm, 1867) | 45 | 377 |
| 10 | *Mithrax aculeatus* (Herbst, 1790) | 45 | 11 |
| 11 | *Omalacantha bicornuta* (Latreille, 1825) | 43 | 1 |
| 12 | *Alpheus armatus* Rathbun, 1901 | 19 | 4 |
| 13 | *Paguristes puncticeps* Benedict, 1901 | 19 | 4 |
| 14 | Callianassid A | 19 | 1 |
| 15 | *Thor amboinensis* (de Man, 1888) | 19 | 0 |
| 16 | *Panulirus argus* (Latreille, 1804) | 18 | 4 |
| 17 | *Macrocoeloma subparellelum* (Stimpson, 1860) | 14 | 0 |
| 18 | *Axiopsis serratifrons* (A. Milne-Edwards, 1873) | 12 | 4 |
| 19 | *Percnon gibbesi* (H. Milne-Edwards, 1853) | 8 | 7 |
| 20 | *Nonala holderi* (Stimpson, 1871) | 7 | 0 |
| 21 | *Petrolisthes galathinus* (Bosc, 1802) | 5 | 38 |
| 22 | *Macrocoeloma diplacanthum* (Stimpson, 1860) | 5 | 0 |
| 23 | Xanthid E | 4 | 0 |
| 24 | *Mithraculus cinctimanus* Stimspon, 1860 | 3 | 3 |
| 25 | *Stenopus hispidus* (Olivier, 1811) | 3 | 0 |
| 26 | *Actaea acantha* (H. Milne-Edwards, 1834) | 2 | 0 |
| 27 | *Ancylomenes pedersoni* (Chace, 1958) | 2 | 0 |
| 28 | *Ratha longimanus* (H. Milne-Edwards, 1834) | 2 | 0 |
| 29 | *Macrocoeloma trispinosum* (Latreille, 1825) | 2 | 0 |
| 30 | *Mithraculus forceps* A. Milne-Edwards, 1875 | 2 | 0 |
| 31 | *Panulirus guttatus* (Latreille, 1804) | 1 | 3 |
| 32 | *Lysmata wurdemanni* (Gibbes, 1850) | 1 | 1 |
| 33 | Anomuran A | 1 | 0 |
| 34 | *Brachycarpus biunguiculatus* (Lucas, 1846) | 1 | 0 |
| 35 | Majoid B | 1 | 0 |
| 36 | Majoid C | 1 | 0 |
| 37 | Majoid D | 1 | 0 |
| 38 | *Neogonodactylus torus* (Manning, 1869) | 1 | 0 |
| 39 | *Pitho lherminieri* (Desbonne in Desbonne & Schramm, 1867) | 1 | 0 |

**Table 1** (*continued*)

| | Species | Bonanza | Limones |
|---|---|---|---|
| 40 | *Pitho mirabilis* (Herbst, 1794) | 1 | 0 |
| 41 | *Podochela macrodera* Stimpson, 1860 | 1 | 0 |
| 42 | *Stenorhynchus seticornis* (Herbst, 1788) | 1 | 0 |
| 43 | Xanthid C | 1 | 0 |
| 44 | Xanthid D | 1 | 0 |
| 45 | *Paguristes cadenati* Forest, 1954 | 0 | 18 |
| 46 | *Phimochirus holthuisi* (Provenzano, 1961) | 0 | 5 |
| 47 | Caridean A | 0 | 3 |
| 48 | *Pachycheles pilosus* (H. Milne Edwards, 1837) | 0 | 3 |
| 49 | *Cinetorhynchus manningi* (Okuno, 1996) | 0 | 2 |
| 50 | Majoid A | 0 | 2 |
| 51 | *Nemausa acuticornis* (Stimpson, 1871) | 0 | 2 |
| 52 | *Damithrax hispidus* (Herbst, 1790) | 0 | 1 |
| 53 | *Maguimithrax spinosissimus* (Lamarck, 1818) | 0 | 1 |
| 54 | *Achelous sebae* (H. Milne Edwards, 1834) | 0 | 1 |
| 55 | *Synalpheus* sp. | 0 | 1 |
| 56 | Xanthid A | 0 | 1 |
| 57 | Xanthid B | 0 | 1 |
| 58 | *Cinetorhynchus rigens* (Gordon, 1936) | X | X |
| 59 | *Metapenaeopsis goodei* (Smith, 1885) | X | X |
| 60 | *Palinurellus gundlachi* von Martens, 1878 | | X |
| 61 | *Parribacus antarcticus* (Lund, 1793) | X | |
| 62 | *Scyllarides aequinoctialis* (Lund, 1793) | X | X |
| 63 | *Carpilius corallinus* (Herbst, 1783) | X | |

of *C. tibicen* did not differ significantly between reefs (Limones: $38.1 \pm 10.2$ ind. 50 m$^{-2}$; Bonanza: $33.4 \pm 10.1$ ind. 50 m$^{-2}$). Two of the most abundant species were recorded on Limones only (*Petrolisthes galathinus*, *Paguristes cadenati*) and one was recorded on Bonanza only (*Paguristes tortugae*) (Fig. 7).

## Microhabitats used by macrocrustaceans

On Limones, the types of microhabitats more commonly occupied by macrocrustaceans were, on descending order, *Millepora* spp., *A. palmata*, coral rubble, dead coral skeletons, and *Agaricia agaricites*, and on Bonanza, coral rubble, dead coral skeletons, macroalgae, *A. agaricites*, and *Millepora* spp. (Fig. 8). Some of these microhabitats constitute components of the benthic community and hence their percent cover was estimated. For example, the average percent cover of *A. palmata* and *Millepora* spp. was higher on Limones (29% and 3.6%, respectively) than on Bonanza (3.5% and 1.9%, respectively). In contrast, the percent cover of fleshy and calcareous macroalgae was higher on Bonanza (32.1% and 15.4%, respectively) than on Limones (19.7% and 5.2%, respectively). However, other types of microhabitat (e.g., coral rubble, relic coral skeletons, sand) were not quantified because they are not components of the benthic community.

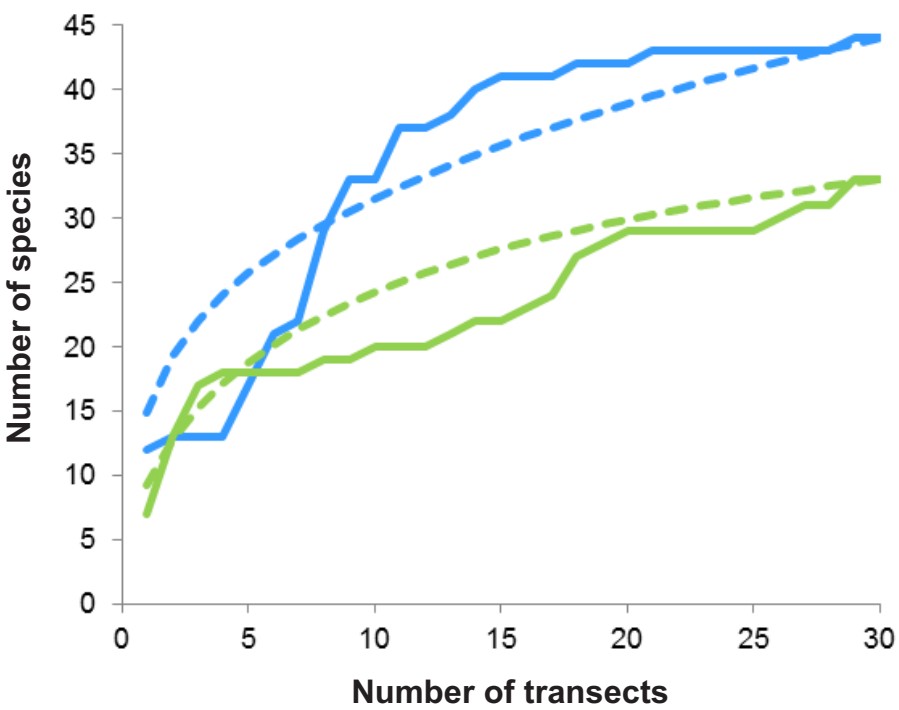

**Figure 5** **Species accumulation and rarefaction curves.** Accumulation curves (continuous lines) and rarefaction curves (dashed lines) for macrocrustacean species recorded in Limones (green lines) and Bonanza (blue lines) reefs. Thirty belt transects, 50 m² each, were sampled on each reef. Rarefaction curves for either reef do not reach an asymptote, indicating the existence of more species.

**Table 2** **Ecological indices for macrocrustaceans by reef.** Mean value (±95% confidence interval) of species richness ($S$), Shannon-Wiener diversity ($H'$), dominance ($D$), and evenness ($J'$) of macrocrustaceans on Bonanza and Limones reefs.

| Ecological index | Bonanza reef | Limones reef |
| --- | --- | --- |
| $S$ | $8.66 \pm 1.18$ | $6.53 \pm 0.71$ |
| $H'$ | $2.07 \pm 0.19$ | $1.54 \pm 0.15$ |
| $D$ | $0.33 \pm 0.04$ | $0.47 \pm 0.05$ |
| $J'$ | $0.69 \pm 0.04$ | $0.58 \pm 0.04$ |

## DISCUSSION

Contrary to our expectations, Bonanza supported a more diverse and abundant macrocrustacean community than Limones, although there were differences between reefs in the percent cover of distinct benthic community components and the types of microhabitats used by macrocrustaceans. Live coral cover (mostly *A. palmata*) was much greater on Limones than on Bonanza, whereas the opposite occurred for fleshy and calcareous macroalgae, and cyanobacterial mats. These results support previous studies concluding that Bonanza has sustained substantial degradation over the past few decades (*Carriquiry et al., 2013*; *Ladd & Collado-Vides, 2013*; *Díaz-Pérez et al., 2016*; *Morillo-Velarde et al., 2018*), whereas Limones is an exceptional site in that it has maintained

2D Stress: 0.24

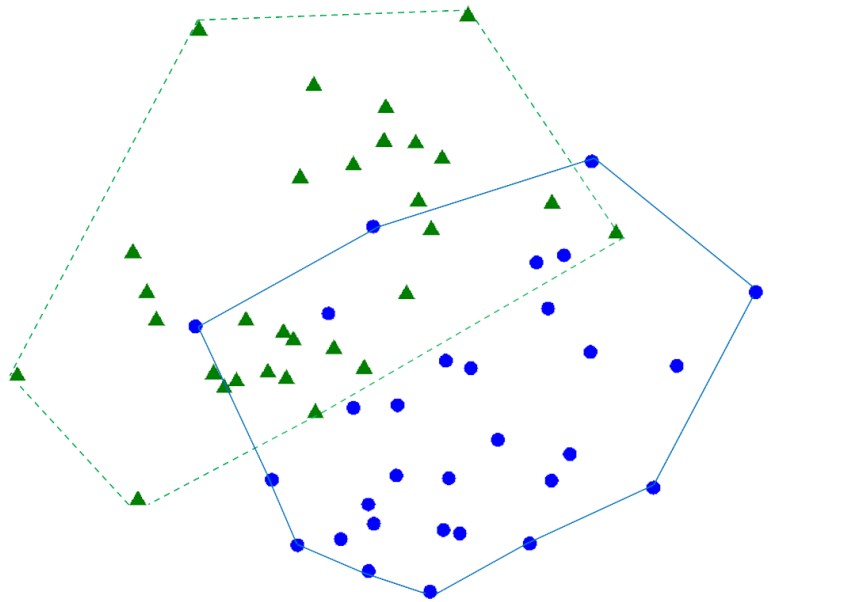

**Figure 6  nMDS ordination.** Non-metric multidimensional (nMDS) ordination of macrocrustacean community structure in samples from Limones reef (green triangles and dashed lines) and Bonanza reef (blue circles and continuous lines), based on species abundances. Each symbol denotes a transect.

healthy populations of *A. palmata* (*Rodríguez-Martínez et al., 2014*). Previously, *Morillo-Velarde et al. (2018)* found a significant difference in rugosity between Limones and Bonanza, but these authors measured this variable in only eight transects over the central part of each reef, where development of *A. palmata* on Limones appears to be greater. In contrast, we did not find a significant difference in the median rugosity between the back-reef zones of these reefs, which could be partially explained by the presence of extensive areas of dead coral skeletons on Bonanza as opposed to the extensive stands of live *A. palmata* on Limones. Thus, although our study was conducted in only two reefs, our results are consistent with studies suggesting that live coral cover is not necessarily a key factor determining the level of structural complexity as long as the reef structure persists (*Lindahl, Öhman & Schelten, 2001*; *Nelson, Kuempel & Altieri, 2016*), i.e., that the relic skeletons (i.e., those left behind after the coral tissue dies) and the structural diversity they create can be important factors determining the diversity and structure of invertebrate communities (*Idjadi & Edmunds, 2006*). However, the wider range in rugosity over Limones, especially of values >2, reflects the patchy presence of more complex substrates interspersed with less rugose substrates. In contrast, the narrower range of values over Bonanza, with few values >2, suggests a lower heterogeneity in substrate rugosity.

Reef invertebrates are highly diverse but hard to sample; in particular, many crustaceans hide deeply in reef crevices or under sediments during the day and only emerge at night to forage (*Glynn & Enochs, 2011*). Therefore, even for conspicuous taxa, visual census methods have several limitations that may result in underestimation of individuals

**Table 3 Similarity measures within and between reefs.** Analysis of similarity percentage (SIMPER) for macrocrustacean assemblages within Limones and Bonanza, and of dissimilarity percentage between reefs.

| Species | AA | AS | Sim/SD | Contrib% | Cum % |
|---|---|---|---|---|---|
| **Limones**. Average similarity: 48.10 | | | | | |
| *Calcinus tibicen* | 2.33 | 21.73 | 4.10 | 45.18 | 45.18 |
| *Mithraculus coryphe* | 1.52 | 12.62 | 2.07 | 26.23 | 71.41 |
| *Domecia acanthophora* | 1.04 | 4.16 | 0.57 | 8.65 | 80.06 |
| *Petrolisthes galathinus* | 0.58 | 2.18 | 0.50 | 4.54 | 84.60 |
| *Teleophrys ruber* | 0.52 | 1.71 | 0.41 | 3.55 | 88.16 |
| *Pagurus brevidactylus* | 0.51 | 1.66 | 0.41 | 3.46 | 91.61 |
| **Bonanza**: Average similarity: 46.90 | | | | | |
| *Mithraculus coryphe* | 2.22 | 15.54 | 3.64 | 33.14 | 33.14 |
| *Calcinus tibicen* | 2.15 | 13.64 | 2.51 | 29.09 | 62.22 |
| *Neogonodactylus oerstedii* | 0.85 | 4.12 | 0.89 | 8.79 | 71.02 |
| *Pagurus brevidactylus* | 0.90 | 3.40 | 0.77 | 7.25 | 78.27 |
| *Mithraculus sculptus* | 0.76 | 2.85 | 0.64 | 6.08 | 84.35 |
| *Omalacantha bicornuta* | 0.52 | 1.29 | 0.45 | 2.75 | 87.10 |
| *Paguristes tortugae* | 0.57 | 1.07 | 0.37 | 2.29 | 89.39 |
| *Teleophrys ruber* | 0.52 | 0.80 | 0.33 | 1.70 | 91.09 |

| Species | Limones AA | Bonanza AA | AD | Dis/SD | Contrib% | Cum % |
|---|---|---|---|---|---|---|
| **Limones and Bonanza**: Average dissimilarity: 58.49 | | | | | | |
| *Domecia acanthophora* | 1.04 | 0.29 | 4.76 | 0.96 | 8.14 | 8.14 |
| *Mithraculus coryphe* | 1.52 | 2.22 | 3.88 | 1.23 | 6.63 | 14.77 |
| *Pagurus brevidactylus* | 0.51 | 0.90 | 3.73 | 1.17 | 6.38 | 21.15 |
| *Calcinus tibicen* | 2.33 | 2.15 | 3.55 | 0.99 | 6.07 | 27.22 |
| *Neogonodactylus oerstedii* | 0.33 | 0.85 | 3.43 | 1.21 | 5.86 | 33.08 |
| *Mithraculus sculptus* | 0.34 | 0.76 | 3.39 | 1.07 | 5.80 | 38.88 |
| *Teleophrys ruber* | 0.52 | 0.52 | 3.10 | 0.97 | 5.30 | 44.18 |
| *Petrolisthes galathinus* | 0.58 | 0.11 | 2.61 | 0.90 | 4.46 | 48.64 |
| *Paguristes tortugae* | 0.00 | 0.57 | 2.28 | 0.72 | 3.90 | 52.54 |

**Notes.**

AA, average abundance; AS, average similarity; Sim/SD, similarity/standard deviation; Contrib%, contribution in %; Cum%, cumulative contribution in %; AD, average dissimilarity; Dis/SD, dissimilarity/standard deviation.

Species are listed in decreasing order of AS within each reef and AD between reefs. Cum.% does not reach 100% in order to facilitate interpretation.

present and sampling error, such as observer variability, characteristics of the target taxa (e.g., crypticity, escape responses), and difficulties imposed by environmental factors (e.g., turbidity, waves, current) (*Lessios, 1996*; *Backus, 2007*). Indeed, a few individuals could only be identified to the superfamily level because they either swam away rapidly (i.e., carideans) or retreated deeply into crevices, or because time or environmental restrictions limited further identification. In addition, because our studied reefs are within a marine protected area, we refrained from collecting but a few individuals for further identification in the laboratory. Despite these limitations, species richness was high on both Limones and Bonanza, as previously reported for other Caribbean reef systems

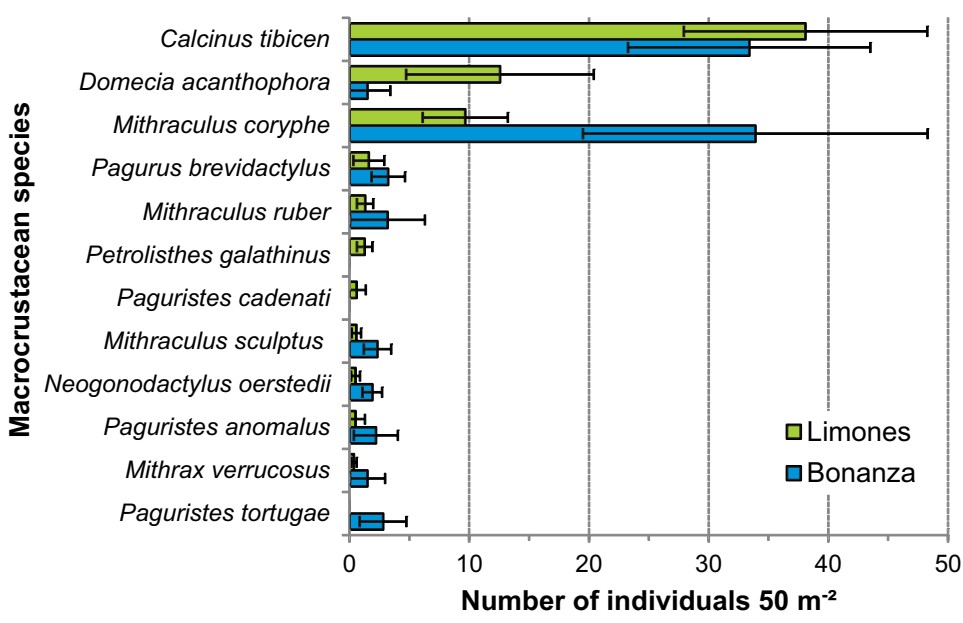

**Figure 7** **Density of macrocrustaceans per reef.** Mean density (number of individuals 50 m$^{-2}$) of the most abundant macrocrustaceans per reef: Limones (green columns) and Bonanza (blue columns). Error bars are 95% confidence intervals.

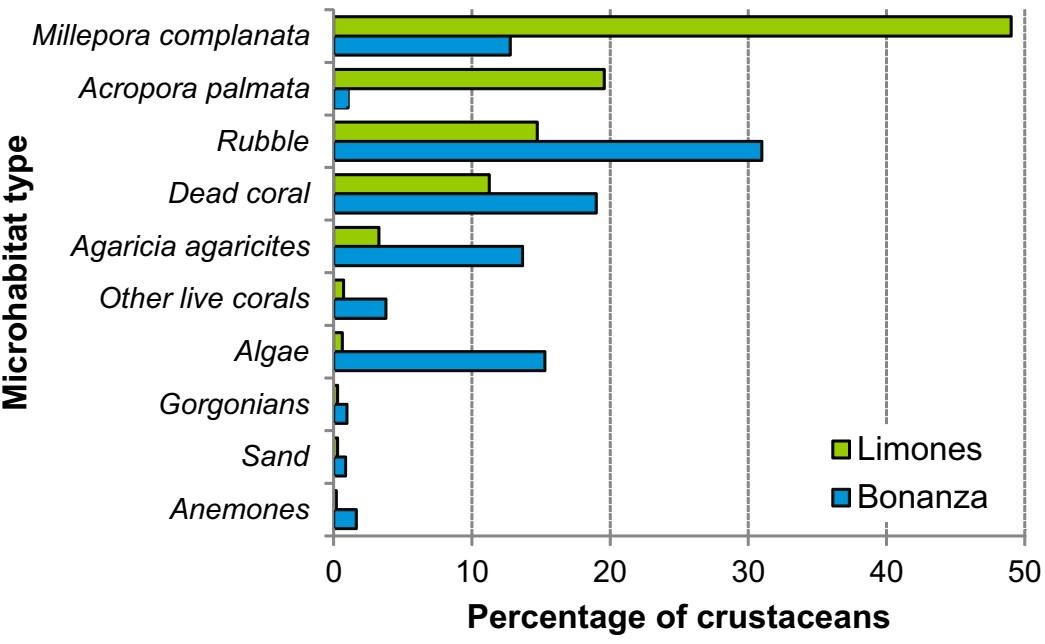

**Figure 8** **Types of microhabitats used by macrocrustaceans.** Comparison of the types of microhabitats used by macrocrustaceans on each reef, Limones (green columns) and Bonanza (blue columns).

(*Reed et al., 1982*; *Martínez-Iglesias & García-Raso, 1999*; *Briones-Fourzán & Lozano-Álvarez, 2002*). However, a more exhaustive sampling would undoubtedly increase the number of macrocrustacean species recorded in these reefs (e.g., Alpheidae and Thalassinidae) as indicated by the rarefaction curves (see Fig. 5).

Most of the species that we observed on both reefs were facultative coral-dwelling crustaceans, i.e., species that are not considered to be fundamentally dependent upon abundant live coral for their local persistence (*Stella et al., 2011*). Although there was some overlap in the macrocrustacean community composition between reefs, most diversity indices were higher on Bonanza except for the dominance index, which was higher on Limones. These results likely reflect a greater heterogeneity of microhabitats on Bonanza, which is characterized by the abundance of relic coral skeletons, coral rubble, and erect fleshy and calcareous macroalgae (*Morillo-Velarde et al., 2018*; the present study), than on Limones, which is characterized by extensive stands of *A. palmata* and short algal turf (*Rodríguez-Martínez et al., 2014*; the present study). The latter would also explain why *D. acanthophora*, a small commensal crab of *Acropora* spp. (*Patton, 1967*), was the main contributor to the dissimilarity between reefs. This species is considered an obligate coral-dwelling crab (*Patton, 1967*), i.e., a species having strong reliance on live corals for food, habitat, and/or recruitment (*Stella et al., 2011*). On the other hand, fleshy and calcareous macroalgae, which were more abundant on Bonanza, offer high quality microhabitats to grazing species (*Roff et al., 2013*) such as majoid crabs of the genera *Mithraculus*, *Mithrax*, *Omalacantha*, and *Maguimithrax*, which use their modified, spooned-shaped chelae to feed on these algae (*Coen, 1988*; *Stachowicz & Hay, 1996*; *Butler IV & Mojica, 2012*). Consequently, majoids were among the most abundant macrocrustaceans on Bonanza, particularly *M. coryphe*, which on this reef was often found in coral rubble overgrown by macroalgae. In Caribbean seagrass habitats, *M. sculptus* outranked *M. coryphe* in abundance (*Carmona-Suárez, 2000*), but similar to our results, *M. coryphe* was the most abundant crab on coralline substrates around an eastern Caribbean island (*García, Hernández & Bolaños, 1998*).

The diogenid *C. tibicen* had a similar abundance as *M. coryphe* on Bonanza, but was the dominant species on Limones. This small hermit crab, which is an omnivorous detritivore (*Hazlett, 1981*), has also been reported as abundant on shallow coral reefs in Panama (*Abele, 1976*), Cuba (*Martínez-Iglesias & García-Raso, 1999*), the Virgin Islands (*Brown & Edmunds, 2013*), and Brazil (*Giraldes, Coelho Filho & Coelho, 2012*). *Brown & Edmunds (2013)* discovered that *C. tibicen* can live commensally on hydrozoans of the genus *Millepora* ("fire corals"). Fire corals were more abundant on Limones than on Bonanza, and many of the individuals of *C. tibicen* that we observed were dwelling on *Millepora* colonies. However, the overall abundance of fire corals was low and we also found *C. tibicen* on virtually all types of microhabitats except for anemones and sand, consistent with *Brown & Edmunds*' (*2013*) conclusion that the association with fire corals is facultative for this crab. In particular, *C. tibicen* was observed in high numbers on relic coral skeletons and coral rubble on both reefs, but especially on Bonanza, where these types of microhabitats abounded.

Specialist species are more vulnerable to disturbances and hence would be expected to be more profoundly affected by coral reef degradation (*Munday, 2004*; *Álvarez Filip et al., 2015*). Based on our results, it would appear that *D. acanthophora*, would be more profoundly affected if Limones underwent an increase in degradation. Indeed, *D. acanthophora* was far more abundant on Limones, where its preferred microhabitat (*A. palmata*) abounded, but we also found it on *Millepora* spp. colonies on both reefs, although proportionally more on Bonanza, suggesting that these small crabs can associate with other sessile invertebrates in the absence of acroporids. For example, *Reed et al. (1982)* recorded *D. acanthophora* in *Oculina* reefs. Interestingly, *Head et al. (2015)* found large numbers of obligate coral-dwelling crabs on dead coral colonies of *Acropora* and *Pocillopora* across five different atolls. These crabs appeared to be explicitly recruiting to or moving to dead coral hosts at certain stages in their life cycle, with no relationship with the abundance of live coral (*Head et al., 2015*).

Our results would appear to confirm that, rather than structural complexity, the variety of microhabitats (i.e., small-scale habitat structure, *Dumas et al., 2013*) is an important factor driving the diversity and abundance of reef-associated crustaceans (*Abele, 1976*; *Head et al., 2015*; *Giraldes et al., 2017*), as is the diversity of mutualistic relationships that these animals can establish with other taxa (e.g., *Patton, 1994*; *Briones-Fourzán et al., 2012*; *Brown & Edmunds, 2013*). For example, *A. palmata* provides habitat for many species, but is very vulnerable to diseases (*Aronson & Precht, 2001*; *Stella et al., 2011*). In Australia, coral colonies displaying a significant reduction in live tissue cover due to partial mortality exhibited an increase in the abundance and richness of small invertebrate species, suggesting that as coral cover is reduced, new microhabitats arise within the colony, allowing other species to occupy new niches (*Stella, Jones & Pratchett, 2010*). Thus, dead *A. palmata* may become important for macrocrustaceans for which the relic coral skeletons and coral rubble are preferred microhabitats. Several studies have already highlighted the importance of dead corals and coral rubble as key microhabitats for reef-dwelling decapod crustaceans (e.g., *Coles, 1980*; *Enochs, 2012*; *Kramer, Bellwood & Bellwood, 2014*; *Head et al., 2015*) and other small invertebrates (*Nelson, Kuempel & Altieri, 2016*). In addition to providing refuge, relic skeletons and coral rubble are typically overgrown by macroalgae, increasing their microhabitat value for herbivorous species (*Roff et al., 2013*).

An increase in the abundance and availability of mobile invertebrates with reef degradation may have positive effects on food web productivity by delaying the loss of other reef components such as fish (*Rogers, Blanchard & Mumby, 2018*), thus potentially giving more time to reef communities to adapt to the new, more unfavorable, conditions. This hypothesis could explain why *Morillo-Velarde et al. (2018)* found a very similar food chain length between Limones and Bonanza reefs despite their contrasting levels of structural and benthic integrity. However, this does not mean that reef-associated crustaceans will benefit from coral reef degradation over the long term, because degraded coral reefs continue to erode over time (*Perry et al., 2012*), eventually reducing the availability of microhabitats with increasing loss of structure and ecosystem functionality (*Przeslawski et al., 2008*; *Head et al., 2015*; *Lozano-Álvarez et al., 2017*), resulting in low productivity over the longer term (*Rogers, Blanchard & Mumby, 2018*). Given the ongoing tendency to increase of coral reef

degradation, future studies should investigate the relative importance of different types of microhabitats at different scales and the occurrence of mutualistic relationships for maintaining diversity and abundance of reef-associated macrocrustaceans.

## CONCLUSIONS

Structural complexity is an important factor driving the diversity and abundance of reef-associated macrocrustaceans, but so is the variety of local microhabitats and mutualistic relationships that these animals can establish with other taxa. We found a greater diversity and abundance of macrocrustaceans in a more degraded coral reef (Bonanza) than in a reef characterized by extensive stands of live *A. palmata* (Limones), but the latter exhibited a higher level of dominance, reflecting the presence in high numbers of a few species that establish mutualistic relationships with *A. palmata* and hydrozoans. On Bonanza, relic skeletons and coral rubble were typically overgrown by macroalgae, thus offering refuge and food to herbivorous macrocrustaceans. However, coral reef degradation continues to increase, making it necessary to investigate the relative importance of different types of microhabitats at different scales and at different levels of degradation for maintaining diversity and abundance of reef-associated macrocrustaceans.

## ACKNOWLEDGEMENTS

The authors wish to acknowledge the invaluable technical assistance provided by C Barradas-Ortiz and F Negrete-Soto throughout the study. Additional assistance in field and/or laboratory work was provided by PS Morillo-Velarde, A Espinosa-Magaña, R Martínez-Calderón, R Candia-Zulbarán, R Muñoz de Cote-Hernández, L Cid-González, N Luviano-Aparicio, and CE Davies. E Escalante-Mancera helped with the use of software. Comments by Darren Brown improved an earlier draft of the manuscript.

### Funding

This work was funded by Universidad Nacional Autónoma de México (Program UNAM-DGAPA-PAPIIT, project IN-205614, granted to Enrique Lozano-Álvarez). Roberto González-Gómez received an MSc scholarship from Consejo Nacional de Ciencia y Tecnología (México), and a complementary scholarship from UNAM-DGAPA-PAPIIT, project IN-205614. The funders had no role in study design, data collection and analysis, decision to publish, or preparation of the manuscript.

### Grant Disclosures

The following grant information was disclosed by the authors:
Universidad Nacional Autónoma de México: (UNAM-DGAPA-PAPIIT), IN-205614.
Consejo Nacional de Ciencia y Tecnología (México).

### Competing Interests

The authors declare there are no competing interests.

## Author Contributions

- Roberto González-Gómez conceived and designed the experiments, performed the experiments, analyzed the data, prepared figures and/or tables, authored or reviewed drafts of the paper, approved the final draft.
- Patricia Briones-Fourzán conceived and designed the experiments, analyzed the data, contributed reagents/materials/analysis tools, prepared figures and/or tables, authored or reviewed drafts of the paper, approved the final draft.
- Lorenzo Álvarez-Filip and Enrique Lozano-Álvarez conceived and designed the experiments, analyzed the data, contributed reagents/materials/analysis tools, authored or reviewed drafts of the paper, approved the final draft.

## Field Study Permissions

The following information was supplied relating to field study approvals (i.e., approving body and any reference numbers):

Field surveys were approved by Comisión Nacional de Acuacultura y Pesca, México (Approval number: PPF/DGOPA-259/14).

## Data Availability

The raw data are provided in a Supplemental File.

## Supplemental Information

Supplemental information for this article can be found online at http://dx.doi.org/10.7717/peerj.4922#supplemental-information.

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
