# Peer review of "Diversity and abundance of conspicuous macrocrustaceans on coral reefs differing in level of degradation"

_PeerJ, doi:10.7717/peerj.4922_

## Round 0.1 · original submission · Major Revisions

Both reviewers have made valuable comments that you should address. In doing so you will greatly increase the value of the report. In particular I think you should pay attention to the commentrs relating to the identification (and possible mis-identification) of species observed and the analysis of reef complexity.

Reviewer 1 ·

Basic reporting

An excellent study with few shortcomings, written in good academic English.

Experimental design

Although Underwater Visual Census techniques are used in a number of studies on Decapoda, they do have a large amount of inherent bias towards highly mobile, conspicuous and easy to identify species. It is therefore not surprising that only one species of Alpheus is recorded. Unquestionably however, several more species of that genus would have been running around on the transect lines, moving under rocks and rubble. It would be good to see some critical text discussing how sampling limitations of a UVC could potentially influence the results.

Validity of the findings

Asides from the above comment on UVC, the findings seem in line with other work.

Additional comments

Although multivariate statistics are highly appropriate for this type of study, I would not think that the calculated diversity indices would be parametric in nature and thus should not be analysed with ANOVA.

It would be good to include sources used to identify the species, was this largely based on UW photography guides or was primary taxonomic literature used? In this regard it appears odd that Mithraculus coryphe was the most abundant species, usually M.sculptus is far more abundant. Perhaps also discuss how the divers were trained in identification, and any bias between them. I would also query why UW photography was not used to get an ID for some taxa, certainly a taxonomist could easily have identified the majority to genus, if not species level of those listed at family level and above. Using "Caridean A" is rather odd as shrimps can easily be identified to family level, and that does raise questions about the ability of the divers to correctly ID taxa. And finally, Cinetorhynchus fasciatus (please not correct spelling of genus) does not occur in the Atlantic ! There are only two known species of that genus in the Atlantic, C.rigens and C.manningi. Again this raises some questions about the accuracy of UVC identifications used herein

Reviewer 2 ·

Basic reporting

This manuscript provides a description of the abundance of crustaceans on two coral reefs that differ in benthic community structure and explores the role of degradation and roughness on crustaceans. Despite decades of studies on coral reefs, studies like this one still are lacking, and reef ecologists continue to focus on fishes and stony corals so that quantitative data on other taxa remain rare. This reason alone makes a compelling reason to publish this study, and indeed the manuscript is well written and very nicely supported by literature. However, I cannot support publication in this format:

1. This study remains an analysis of two reefs at a single time point using 30 transects at each site. The authors need to make a stronger case that this study has general value. I am confident this can be done, but there seems little attempt to do this in the current version. Even the notion that this is a contrast of a degraded and less degraded reef is ambiguous, as no data to this effect are presented, although some citations to this effect are added at line 127. It would greatly strengthen this study to add a little time series context to Fig. 2.

2. The analysis of benthic roughness by the “HAS” index is exceptionally weak, and even the application of the better-known rugosity index is done in a non standard way. This is surprising since one of the authors has published a very nice review on reef roughness. At a time when ecologists are using increasingly sophisticated tools to quantify benthic roughness, the HAS method, which is just a qualitative ranking, is not an acceptable tool. The rugosity method was conducted with un-weighted transect which would not conform to the benthos (this method is usually done with a weighted chain). The same tape was used to measure benthic community structure, and the low resolution of this method (10 points 10 cm apart down a 10 m line) surely contributed to the high variance in the benthic data (Fig. 2). Taken together, it is hardly surprising that the reefs did not differ in roughness.

In general, given the simplicity of this study and the limited generality of the outcome, this would need to be greatly trimmed in length to justify publication.

Experimental design

Weak

Validity of the findings

Very narrow -- two reefs one time point. Generality needs to be established.

---

## Round 0.2 · accepted · Accept

You have done an excellent job of responding to the reviewer's comments in particular highlighting the fact that you were looking at conspicuous species and dealing with the few questionable identifications.

# Reviewer 1 ·

Basic reporting

no further comment

Experimental design

no further comment

Validity of the findings

no further comment

Additional comments

The authors have adequately addressed my review comments, and I thus recommend publication.